# Fresh versus Frozen Embryo Transfer in In Vitro Fertilization/Intracytoplasmic Sperm Injection Cycles: A Systematic Review and Meta-Analysis of Neonatal Outcomes

**DOI:** 10.3390/medicina60081373

**Published:** 2024-08-22

**Authors:** Raluca Tocariu, Lucia Elena Niculae, Alexandru Ștefan Niculae, Andreea Carp-Velișcu, Elvira Brătilă

**Affiliations:** 1Department of Obstetrics and Gynecology, Carol Davila University of Medicine and Pharmacy, 050474 Bucharest, Romania; andreea_veliscu@yahoo.com (A.C.-V.); elvirabarbulea@gmail.com (E.B.); 2Clinical Hospital of Obstetrics and Gynecology “Prof. Dr. Panait Sârbu”, 060251 Bucharest, Romania; 3Mother and Child Department, University of Medicine and Pharmacy “Iuliu Hațieganu”, 400347 Cluj-Napoca, Romania; alexandru-stefan.niculae@rez.umfcd.ro

**Keywords:** in vitro fertilization, embryo transfer, adverse birth outcomes, neonatal prematurity

## Abstract

*Background and Objectives*: Although considerable research has been devoted to examining the distinctions between fresh and frozen embryo transfer regarding obstetric outcomes and rates of pregnancy success, there is still a scarcity of thorough analyses that specifically examine neonatal outcomes. The objective of our study was to provide an in-depth analysis of neonatal outcomes that occur after the transfer of fresh and frozen embryos (ET vs. FET) in IVF/ICSI cycles. *Materials and Methods*: Multiple databases (PubMed/MEDLINE, Cochrane Library, Web of Science, Wiley, Scopus, Ovid and Science Direct) were searched from January 1980 to February 2024. Two reviewers conducted the article identification and data extraction, meeting inclusion and exclusion criteria. The methodological quality was evaluated using the Newcastle–Ottawa Scale (NOS) or the revised Cochrane Risk of Bias Tool. The meta-analysis was performed using RevMan 5.4. *Results*: Twenty studies, including 171,481 participants in total, were subjected to qualitative and quantitative analyses. A significant increase in preterm birth rates was noted with fresh embryo transfer compared to FET in the overall IVF/ICSI population (OR 1.26, 95% CI 1.18–1.35, *p* < 0.00001), as well as greater odds of a low birth weight (OR 1.37, 95% CI 1.27–1.48, *p* < 0.00001) and small-for-gestational-age infants in this group (OR 1.81, 95% CI 1.63–2.00, *p* < 0.00001). In contrast, frozen embryo transfer can result in macrosomic (OR 0.59, 95% CI 0.54–0.65, *p* < 0.00001) or large-for-gestational-age infants (OR 0.64, 95% CI 0.60–0.69, *p* < 0.00001). No significant difference was observed regarding congenital malformations or neonatal death rates. *Conclusions*: This systematic review confirmed that singleton babies conceived by frozen embryo transfer are at lower risk of preterm delivery, low birthweight and being small for gestational age than their counterparts conceived by fresh embryo transfer. The data support embryo cryopreservation but suggest that elective freezing should be limited to cases with a proven indication or within the framework of a clinical study.

## 1. Introduction

Since the birth of Louise Brown in 1978 [1,2], more than 10 million babies have been born worldwide as a result of in vitro fertilization (IVF) and intracytoplasmic sperm injection (ICSI) [3]. Conventionally, assisted reproductive technology (ART) consists of fresh embryo transfer (ET) directly after ovarian hyperstimulation. With the refinement of technology in recent years, the number of thawed frozen embryo transfers (FETs) has increased as frozen embryos can be used at a later time, when the ovaries are not stimulated, thereby reducing the risk of ovarian hyperstimulation syndrome (OHSS, an overreaction to fertility drugs) [4].

Although considerable research has been devoted to examining the distinctions between fresh and frozen embryo transfer regarding obstetric outcomes and the rates of pregnancy success, there is still a scarcity of thorough analyses that specifically examine neonatal outcomes. Neonatal health is critical in determining the success and safety of ART procedures because it directly impacts offspring long-term well-being and has implications for parental satisfaction and healthcare resource allocation [5].

The aim of this systematic review and meta-analysis was to provide a thorough examination of the current body of literature regarding neonatal outcomes that occur after the transfer of fresh and frozen embryos in IVF/ICSI cycles, covering factors such as birthweight, gestational age, congenital anomalies, and neonatal mortality.

## 2. Materials and Methods

The PRISMA guidelines for systematic reviews were followed [6,7].

The protocol was registered at PROSPERO (CRD42024524191).

### 2.1. Participants

#### 2.1.1. Inclusion Criteria

Women aged between 20 and 50 years old, with known causes of infertility and pregnancies conceived following autologous fresh or frozen embryo transfer, were included.

#### 2.1.2. Exclusion Criteria

Use of donor gametes;Preimplantation genetic screening or diagnosis;Twins, triplets or vanishing twins;Ectopic pregnancies;Surrogate pregnancies;Patients with significant pregestational disease;Unknown cause of infertility;Low-quality embryo transfers (according to the Istanbul consensus on embryo assessment) [8];Incomplete medical data.

### 2.2. Outcomes

#### 2.2.1. Primary Outcome

The primary objective was to determine whether a policy of freezing embryos, followed by thawed frozen-embryo transfer results in a lower incidence of preterm birth (under 37 weeks of gestational age at delivery) than the current policy of transferring fresh embryos.

#### 2.2.2. Secondary Outcomes

The secondary outcome measures were small or large for gestational age, low birthweight, macrosomia, congenital malformations and neonatal death.

They are defined as follows:Small for gestational age (SGA)—birthweight less than the 10th percentile or less than 2 standard deviations for gestational age, according to international child growth standards.Large for gestational age (LGA)—birthweight greater than the 90th percentile or more than 2 standard deviations for gestational age, according to international child growth standards.Low birthweight (LBW)—weighing less than 2500 grams at birth.Macrosomia—weighing more than 4000 grams at birth.Congenital malformations—any congenital anomaly found in the newbornNeonatal death—death occurring after birth and before day 29.

### 2.3. Types of Studies

Randomized controlled trials and cohort studies were performed.

### 2.4. Search Strategy

PubMed/MEDLINE, Cochrane Library, Web of Science, Wiley, Scopus, Ovid and Science Direct were searched for articles published between January 1980 and February 2024. Additionally, the website www.ClinicalTrials.gov was also checked for ongoing randomized clinical trials that could be included. Finally, after identifying the relevant studies, we performed a manual search through their reference lists to find other potentially relevant trials.

The search was conducted using special vocabulary and keywords, as follows:First, we performed a simple search, such as “in vitro fertilization fresh versus frozen neonatal outcome”;Next, we added age and study filters, as well as keywords (“preterm birth”, “prematurity”, “SGA”, “LGA”, “macrosomia”, “low birth”, “congenital”, “malformations”, “neonatal death”), which we combined using AND, OR and NOT connectors;Furthermore, we used Clinical Queries for Therapy and Prognosis to search for “in vitro fertilization fresh versus frozen embryo neonatal outcome”.

The steps mentioned above are examples of the search strategy used for the PubMed/MEDLINE database. For other databases, every step has been adapted to their respective available filters.

### 2.5. Study Selection, Data Extraction and Risk of Bias Assessment

After removing the duplicates, one author reviewed the titles and abstracts of the identified trials. The studies that satisfied the inclusion criteria were fully read by two independent authors. Any disagreement regarding the trials to be included in the systematic review and the meta-analysis was resolved by discussion.

One author extracted the data from the relevant studies using a standardized table, while the second author checked the accuracy of this process. Information regarding the study design, participants, comparison group and outcomes was included.

The methodological quality of the studies was evaluated using the Newcastle–Ottawa Scale (NOS) for cohort studies [9,10] or the latest version of the Cochrane Risk of Bias Tool for randomized controlled trials [11]. On the one hand, the risk of bias is classified on the NOS from 0 to 9 stars, for which a higher score indicates better quality (8 or 9, high; 6 or 7, moderate; less than 5, low quality). On the other hand, the latter consists of several questions with five possible answers (Yes; Probably Yes; No; Probably No; No Information) that must be followed by an argument providing support. Ultimately, the authors chose the overall risk of bias from three possible categories: low risk of bias, some concerns and high risk of bias. The clinical trials with a low risk of bias are considered to be of high methodological quality, while the other two categories signal out trials with a low methodological quality.

### 2.6. Data Analysis

The meta-analyses were performed using the RevMan 5.4 software package [12]. In case of persistent heterogeneity (I^2^ > 50%), we opted for the random-effects model for continuous outcome data. We chose the fixed-effects model for studies with minimal heterogeneity. The effect of the interventions was measured using odds ratios with 95% confidence intervals. We included only adjusted data when available.

## 3. Results

### 3.1. Search Results

We identified 1409 trials that analyzed neonatal outcomes following fresh versus frozen FIV/ICSI embryo transfers. After removing the duplicates, the titles and abstracts of the remaining studies were filtered to determine whether they satisfied the inclusion criteria. Ultimately, 67 trials were chosen for full-text retrieval, 20 of which were included in the systematic review. 

Considering that one study encompasses more than half of the total number of patients and three others have various confounding variables, 16 out of 20 studies were included in the meta-analysis.

The selection process, as well as explanations regarding the excluded studies, is presented in Figure 1.

### 3.2. Included Studies and Their Risk of Bias

The trials included were conducted over a variable period of time, from 2 to 20 years, and included 171,481 participants in total. All the details regarding the study design, duration, participants and outcomes are included in Table 1.

The risk of bias assessment revealed very high methodological quality. While sixteen cohort studies received a maximum score on the NOS, the other two achieved 8 out of 9 stars, as they included solely women of advanced maternal age, who are prone to preterm delivery [33]. Regarding randomized controlled trials, we added two non-blinded, parallel studies that, according to the Cochrane Risk of Bias Tool questionnaires, had a low and moderate risk of bias. This was compounded by the non-adherence to the allocated intervention of up to 31% in the latter study.

We assessed publication bias using funnel plots for each outcome, testing for asymmetry with Egger’s and Begg’s Tests. The non-significant *p*-values indicate symmetry, suggesting no substantial publication bias in our meta-analysis (results attached as Appendix A).

### 3.3. Effect of Fresh Versus Frozen Embryo Transfer (ET vs. FET) on the Incidence of Prematurity

Fifteen studies reported the cumulative rates of preterm delivery, including 35,625 infants in the fresh embryo transfer group and 19,021 infants in the frozen embryo transfer group. The overall odds ratio was 1.26 (95% CI 1.18, 1.35, Figure 2).

To reduce heterogeneity (I^2^ = 85%, Chi^2^ = 92.01), we performed a new meta-analysis using a random-effects model (Figure 3). However, we found a similar statistically significant result, hence concluding that fresh ET may increase the incidence of preterm birth.

### 3.4. Effect of Fresh versus Frozen Embryo Transfer (ET vs. FET) and Low Birthweight

We pooled the data from fourteen studies addressing the incidence of low birthweight (LBW) and generated a forest plot using a fixed-effects model (Figure 4). Compared to singletons born after frozen embryo transfer, children in the ET group had greater odds of weighing less than 2500 grams at birth (OR = 1.37 with a very narrow confidence interval and strongly significant *p*-value).

### 3.5. Effect of Fresh versus Frozen Embryo Transfer (ET vs. FET) and Macrosomia

The occurrence of macrosomia was more frequent within the FET group, according to six research papers included in the meta-analysis (OR = 0.59, 95% CI 0.54, 0.65, *p* < 0.00001—Figure 5).

### 3.6. Effect of Fresh versus Frozen Embryo Transfer (ET vs. FET) and Small-for-Gestational-Age (SGA) Infants

Eleven studies published between 2010 and 2023, with a cohort of 44,255 singletons, raised concerns regarding birthweight under the 10th percentile in offspring conceived by fresh embryo transfer. Compared to infants in the frozen embryo transfer group, infants born after ET were significantly smaller for gestational age (OR 1.81, 95% CI 1.63, 2.00, with low heterogeneity and *p* < 0.05—Figure 6).

### 3.7. Effect of Fresh versus Frozen Embryo Transfer (ET vs. FET) and Large for Gestational Age (LGA) Infants

A total of nine studies reported this neonatal outcome. Similarly to infants with macrosomia, infants in the FET group had greater odds of being born large for gestational age than those born after fresh embryo transfer (OR = 0.64, with a very narrow confidence interval and almost null heterogeneity—Figure 7).

### 3.8. Effect of Fresh versus Frozen Embryo Transfer (ET vs. FET) and Congenital Malformations

No significant difference was found for the incidence of congenital malformations between the vitrified and fresh blastocyst groups (see Figure 8).

### 3.9. Effect of Fresh versus Frozen Embryo Transfer (ET vs. FET) and Neonatal Death

Only two studies [13,19] reported the cumulative rates of neonatal death, and therefore, pooling was not possible. Based on their conclusions, neonatal death rates were comparable between the ET and FET groups.

## 4. Discussion

### 4.1. Principal Findings

This is a definitive, updated systematic review on a key topic in assisted reproduction. Our findings suggest that offspring conceived by fresh embryo transfer are more prone to preterm delivery, with a low birthweight or birthweight less than the 10th percentile for gestational age. On the other hand, frozen embryo transfer can result in a macrosomic or large-for-gestational-age infant. No significant difference was observed regarding congenital malformations or neonatal death rates between the fresh ET group and the FET group.

### 4.2. Comparison with Other Studies

Our results have been consistent in terms of direction and magnitude of effect over several years, corroborating the findings of recent systematic reviews [4,34,35,36]. However, a meta-analysis published in 2020 highlighted that the risks of congenital anomalies and chromosomal aberrations in newborns associated with fresh embryo transfer showed an absolute increase when compared with frozen embryo transfer (RR 1.09 95% CI 1.02, 1.17, *p* = 0.009, heterogeneity I^2^ = 0%) [37].

### 4.3. Explanation of Results

Endometrial preparation is a crucial stage in FET cycles, and several protocols are available, including a true natural cycle with spontaneous ovulation, a modified natural cycle with human chorionic gonadotrophin (hCG) to trigger ovulation, a hormone replacement therapy (HRT) cycle with or without gonadotropin-releasing hormone agonist (GnRH-a) downregulation, and an ovarian stimulation cycle with or without letrozole. However, there is no consensus on the optimal endometrial preparation protocol for FET [38,39]. In recent years, it has been suggested that artificial endometrial preparation is associated with increased birthweight. In this review, seven of the included studies used only artificial cycles for frozen embryo transfer [14,18,20,22,26,27,29].

Another observation regarding FET cycles was noted by Hwang et al. in their large retrospective cohort of 14,491 infants [21]. Their novel finding was the recurrence of infectious diseases and greater odds for respiratory and neurologic conditions compared to those conceived after fresh embryo transfer. Considering that neonatal morbidity is potentially related to traumatic events in LGA or macrosomia cases [40], the aforementioned remarks regarding infants in the FET group could be secondary to the high incidence of LGA.

### 4.4. Strengths and Limitations

Our metanalyses include a larger number of studies with a far larger number of individual participants (more than six trials and 17 thousand neonates for each forest plot) than previous publications that have evaluated neonatal outcomes after IVF procedures [4]. Also, there are subtle yet significant differences in the way that previous authors have constructed their IVF categories (“conventional” and “freeze-all”) versus our comparison of “fresh embryo transfer” (ET) versus “frozen embryo transfer” (FET). Regarding neonatal outcomes, a combined outcome of perinatal and neonatal death was previously reported, but this does not differentiate between the very different epidemiological characteristics of perinatal deaths (such as stillbirths) and neonatal deaths due to pathology treated in the neonatal intensive care unit.

At the end of our literature search, we chose not to include four studies in our meta-analysis.

First, in 2016, Maheshwari et al. published an extensive retrospective cohort study, following a total of 112,432 singleton live births after IVF/ICSI cycles [23]. Based on data extracted between 1991 and 2012, they concluded that pregnancies resulting from frozen embryo transfers have an increased risk of macrosomia and a decreased risk of LBW, with no difference regarding preterm birth or congenital malformations. This is the largest study in our systematic review, involving more than 65% of participants, but it has one major limitation. According to the authors, they were unable to identify pregnant women included more than once in their dataset; hence, they chose to report 99.5% confidence intervals instead of the traditional value of 95%, while the statistical significance was set at 0.005. Owing to possibly overlapping categories, significant sample sizes and particular reporting formats, we opted to omit this trial as it could have influenced our findings.

Second, we identified subtle, inexplicable shifts regarding the number of participants and the assessment of various outcomes in two trials included in the systematic review [25,30]. The lack of data homogeneity among the limited study populations prompted the decision to exclude the retrospective cohorts from the meta-analysis.

Third, after evaluating the risk of bias using the Cochrane Risk of Bias Tool, we identified one out of two randomized clinical trials with lower methodological quality of evidence, as a large number of patients (up to 31%), especially in the freeze-all arm, were lost due to non-adherence [24].

Given the reasons stated above, we chose not to perform a sensitivity analysis. We consider that homogenous data, low heterogeneity (demonstrated by low to null Chi^2^ and I^2^), a lack of variability in the study quality (by including uniformly high-quality studies) and following a predefined protocol without deviations and variations in methodology (prospectively registered in PROSPERO) strengthen the validity of our conclusions.

However, there are certain limitations to this review. We combined both major and minor fetal abnormalities, as these were not available as separate data for half of the studies. Nevertheless, in trials with a distinctive demarcation line between categories, a significant proportion of the detected congenital malformations were classified as serious, causing functional impairment or need. Additionally, as there are few randomized controlled trials reporting perinatal outcomes in singleton pregnancies, this review is limited to data from observational studies that are, for the most part, constrained by their retrospective nature. Finally, bias may have been introduced as data not published as full-text articles and in languages other than English were excluded from the meta-analysis.

## 5. Conclusions

This systematic review confirmed that singleton babies conceived by frozen embryo transfer are at lower risk of preterm delivery, low birthweight and being small for gestational age than their counterparts conceived by fresh embryo transfer. No significant difference was observed regarding congenital malformations or neonatal mortality between the two groups.

In light of the findings from this meta-analysis and corroborating studies in the field, we strongly recommend that clinicians personalize embryo transfer strategies to optimize neonatal outcomes and maternal health. Given the evidence favoring frozen embryo transfer (FET) in reducing risks such as preterm birth and low birth weight, it may be beneficial to preferentially recommend FET in cases without contraindications. It is crucial, however, to consider individual patient profiles, including age, infertility etiology and previous ART outcomes when devising treatment plans. Clinicians should also stay updated with advancements in embryo freezing techniques and the development of new cryopreservatives that may further enhance the success rates of FET. Incorporating patient education about the potential benefits and limitations of each embryo transfer option will facilitate shared decision making and help align treatment choices with patient expectations and clinical realities.

The future of assisted reproductive technology promises substantial advancements as research continues to uncover the intricate dynamics of embryo transfer and its implications for both immediate and long-term neonatal outcomes. An area ripe for exploration is the molecular and cellular impact of the freeze–thaw cycle on embryos and how this interacts with the endometrial environment during implantation [38]. Understanding these interactions in greater depth could lead to innovations in embryo handling and transfer techniques that minimize cellular stress and optimize implantation success.

Furthermore, longitudinal studies tracking the health of children born via ART are essential to assess the developmental and health trajectories of this population. Such research should also extend to comparing different protocols within ART to ascertain best practices. With the increasing utilization of genomic technologies and bioinformatics, the potential to personalize ART treatments based on the genetic and epigenetic characteristics of patients and embryos could dramatically enhance treatment efficacy and safety [41,42].

In conclusion, as the field of ART evolves, continuous research, technological development and clinical refinement are necessary to ensure that the benefits of these advancements reach the patients. By fostering a robust dialog between researchers, clinicians and patients, the ART community can strive towards a future where every individual has access to safe, effective and personalized reproductive care.

## Figures and Tables

**Figure 1 medicina-60-01373-f001:**
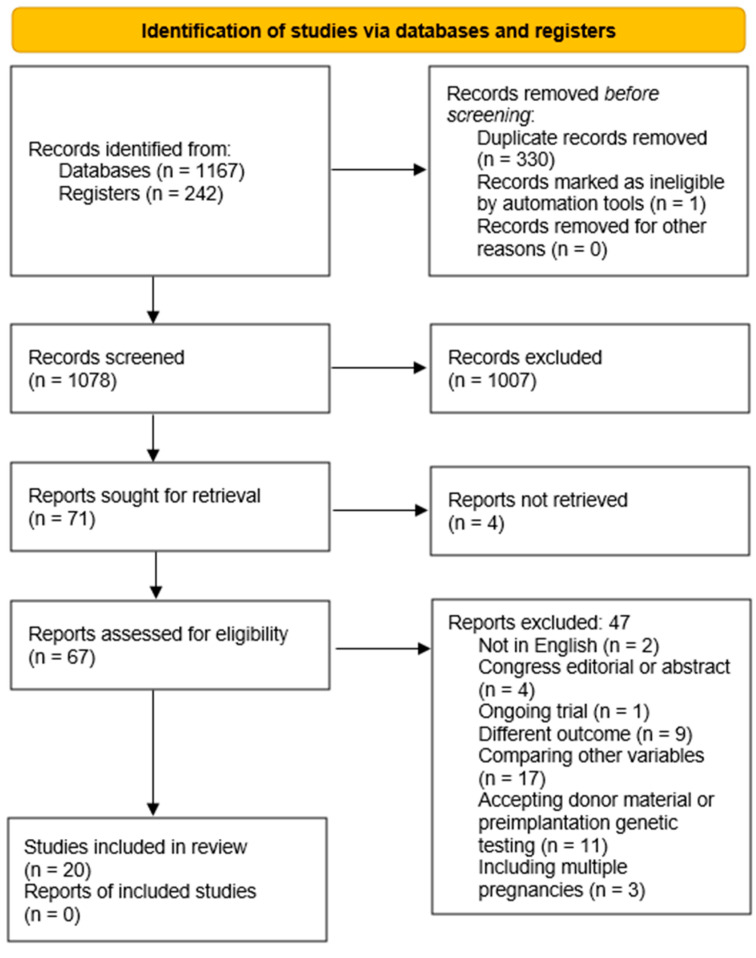
The PRISMA flow diagram completed with the search results.

**Figure 2 medicina-60-01373-f002:**
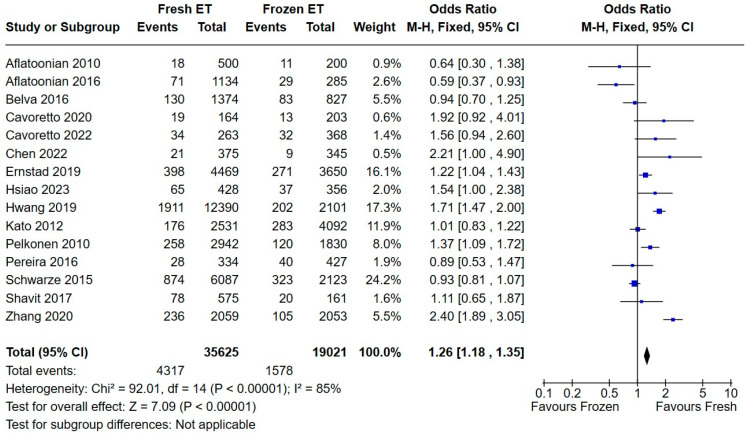
Effect of fresh versus frozen embryo transfer (ET vs. FET) on the incidence of prematurity—fixed-effects model [13,14,15,16,17,18,19,20,21,22,26,27,28,29,32].

**Figure 3 medicina-60-01373-f003:**
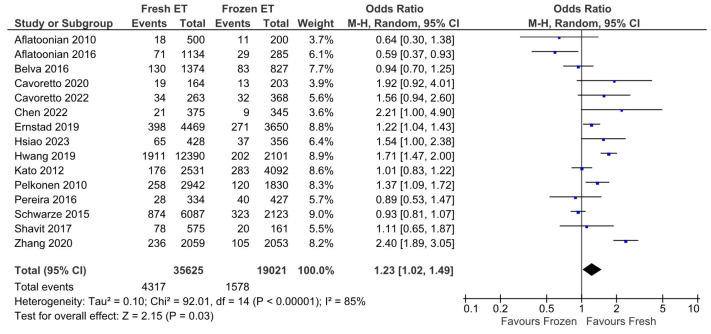
Effect of fresh versus frozen embryo transfer (ET vs. FET) on the incidence of prematurity—random-effects model [13,14,15,16,17,18,19,20,21,22,26,27,28,29,32].

**Figure 4 medicina-60-01373-f004:**
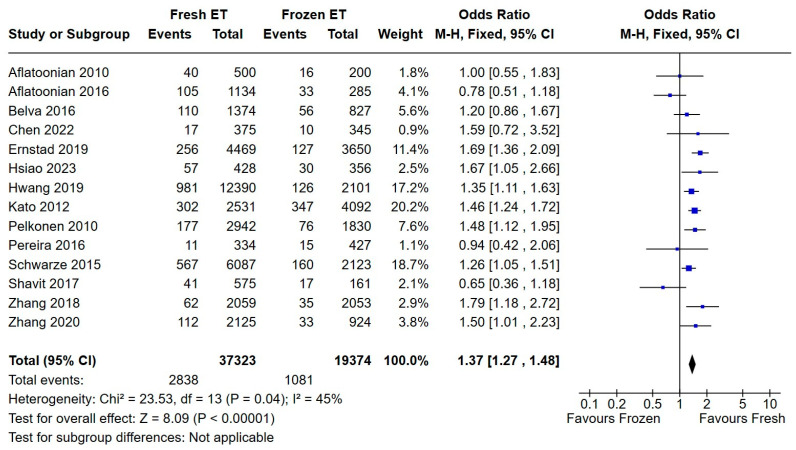
Fresh versus frozen embryo transfer (ET vs. FET) and LBW [13,14,15,18,19,20,21,22,26,27,28,29,31,32].

**Figure 5 medicina-60-01373-f005:**
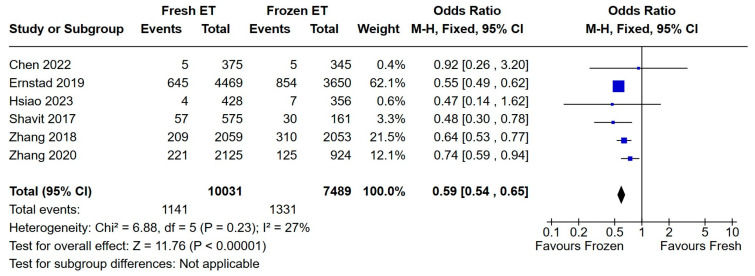
Fresh versus frozen embryo transfer (ET vs. FET) and macrosomia [18,19,20,29,31,32].

**Figure 6 medicina-60-01373-f006:**
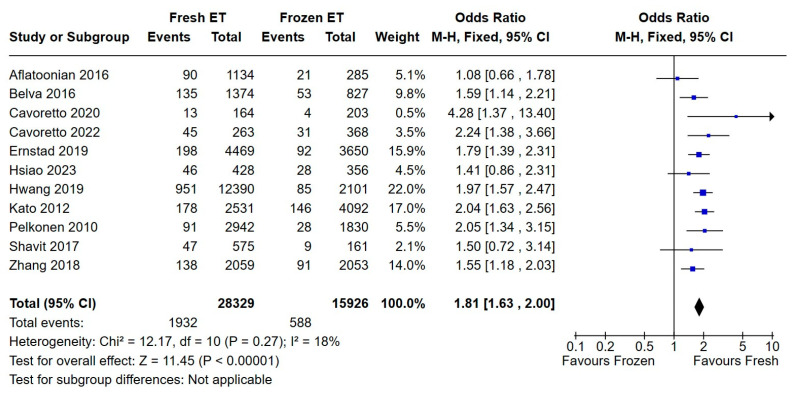
Fresh versus frozen embryo transfer (ET vs. FET) and SGA [14,15,16,17,19,20,21,22,26,29,31].

**Figure 7 medicina-60-01373-f007:**
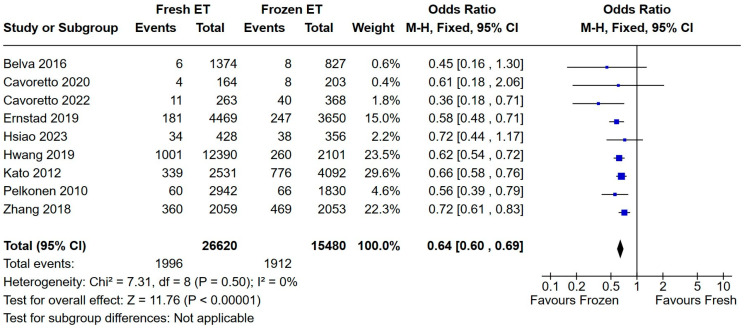
Fresh versus frozen embryo transfer (ET vs. FET) and LGA [15,16,17,19,20,21,22,26,31].

**Figure 8 medicina-60-01373-f008:**
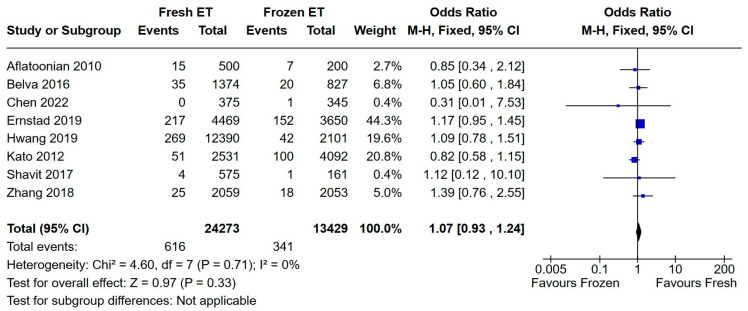
Fresh versus frozen embryo transfer (ET vs. FET) and congenital malformations [13,15,18,19,21,22,29,31].

**Table 1 medicina-60-01373-t001:** Characteristics of the included studies.

Author, Country, Year of Publication	Type of Study, Duration	Population	Neonatal Outcome	Risk of Bias Assessment
Aflatoonian et al. Iran, 2010 [13]	Prospective cohort 2 years	ET (*n* = 500) FET (*n* = 200)	No significant difference between the two groups regarding singleton pregnancies and preterm birth, LBW, neonatal death or congenital malformations.	Maximum score on the NOS
Aflatoonian et al. Iran, 2016 [14]	Prospective cohort 4 years	ET (*n* = 1134) FET (*n* = 285)	No significant difference between the two groups regarding singleton pregnancies and SGA or LBW. FET significantly increases the risk of prematurity in singleton pregnancies, compared to ET (OR 1.65 [1.03–2.66]; *p* = 0.037).	Maximum score on the NOS
Belva et al. Belgium, 2016 [15]	Prospective cohort 5 years	ET (*n* = 1374) FET (*n* = 827)	FET singletons are less likely to be born SGA (OR 0.55 [0.34–0.9]; *p* = 0.005). Otherwise, there were comparable neonatal outcomes between the two groups regarding preterm birth, LBW, LGA or congenital malformations.	Maximum score on the NOS
Cavoretto et al. Italy, 2020 [16]	Prospective cohort 3 years	ET (*n* = 164) FET (*n* = 203)	ET significantly increases the incidence of SGA singletons (OR 4.28 [1.37–13.4]; *p* = 0.008). No significant difference between the two groups regarding singleton pregnancies and LGA or preterm birth.	Maximum score on the NOS
Cavoretto et al. Italy, 2022 [17]	Prospective cohort 5 years	ET (*n* = 263) FET (*n* = 368)	ET significantly increases the incidence of SGA singletons (OR 2.24 [1.38–3.66]; *p* < 0.001). ET singletons are less likely to be born LGA compared to FET singletons (OR 0.36 [0.18–0.71]; *p* = 0.002). No significant difference between the two groups regarding singleton pregnancies and preterm birth.	Maximum score on the NOS
Chen et al. China, 2022 [18]	Retrospective cohort 4 years, 5 months	ET (*n* = 375) FET (*n* = 345)	ET is associated with a higher risk of preterm birth (OR 2.21 [1–4.9]; *p* = 0.046). No significant difference between the two groups regarding singleton pregnancies and macrosomia, LBW or congenital malformations.	NOS score: S ★★★★ C ★★ O ★★★
Ernstad et al. Sweden, 2019 [19]	Prospective cohort 13 years	ET (*n* = 4469) FET (*n* = 3650)	Transfer of vitrified blastocysts was associated with a lower risk of LBW (OR 0.57 [0.44–0.74]) and SGA (OR 0.58 [0.44–0.78]), yet a higher risk of macrosomia (OR 1.77 [1.35–2.31]) and LGA (OR 1.48 [1.18–1.84]). No significant difference between the two groups regarding singleton pregnancies and preterm birth, neonatal death or congenital malformations.	Maximum score on the NOS
Hsiao et al. Taiwan, 2023 [20]	Retrospective cohort 12 years	ET (*n* = 428) FET (*n* = 356)	Singletons conceived via ET were at a higher risk of preterm delivery (OR 1.54 [1–2.38]; *p* = 0.047) and LBW (OR 1.67 [1.05–2.66]; *p* = 0.028). No significant difference between the two groups regarding singleton pregnancies and macrosomia, SGA or LGA.	Maximum score on the NOS
Hwang et al. United States of America, 2019 [21]	Retrospective cohort 9 years, 6 months	ET (*n* = 12,390) FET (*n* = 2101)	Infants conceived by FET have higher odds of LGA (OR 1.47 [1.26–1.7]), with lower odds of SGA (OR 0.56 [0.44–0.7]) and LBW (OR 0.72 [0.59–0.88]). There was no significant difference regarding preterm birth or congenital malformations.	Maximum score on the NOS
Kato et al. Japan, 2012 [22]	Retrospective cohort 3 years	ET (*n* = 2531) FET (*n* = 4092)	Infants conceived by FET have lower odds of SGA (OR 0.43 [0.33–0.56]) and LBW (OR 0.65 [0.53–0.79]). There was no significant difference regarding preterm birth, LGA or congenital malformations.	Maximum score on the NOS
Maheshwari et al. United Kingdom, 2016 [23]	Retrospective cohort 20 years	ET (*n* = 95,911) FET (*n* = 16,521)	Singleton pregnancies following FET are associated with a lower risk of LBW and a higher risk of macrosomia. There was no significant difference regarding preterm birth and congenital malformations. * There may be overlapping in selected categories. The authors selected a personalized reporting of data (different confidence interval and *p*-value, with adjusted risk ratio).	Maximum score on the NOS
Maheshwari et al. United Kingdom, 2022 [24]	Randomized controlled trial, non-blinded, parallel 3 years, 2 months	ET (*n* = 309) FET (*n* = 307)	There was no significant difference regarding singleton pregnancies and preterm delivery, SGA, LGA, LBW, macrosomia, neonatal death or congenital malformations. * The authors selected a personalized reporting of data (different confidence interval, with unadjusted risk ratio).	Moderate risk of bias using the Cochrane Risk of Bias Tool
Ozgur et al. Turkey, 2015 [25]	Retrospective cohort 2 years	ET (*n* = 176) FET (*n* = 116)	There was no significant difference regarding singleton pregnancies and preterm birth or LBW. * Data were reported using the risk ratio. The cohort is slightly different between measurements.	Maximum score on the NOS
Pelkonen et al. Finland, 2010 [26]	Retrospective cohort 11 years	ET (*n* = 2942) FET (*n* = 1830)	The FET group has significantly decreased risks of preterm birth (OR 0.83 [0.71–0.97]), LBW (OR 0.74 [0.62–0.88]) and SGA (OR 0.63 [0.49–0.83]), but an increased risk of LGA (OR 1.7 [1.21–2.40]) in comparison with the ET group.	Maximum score on the NOS
Pereira et al. United States of America, 2016 [27]	Retrospective cohort 3 years, 9 months	ET (*n* = 334) FET (*n* = 427)	There was no significant difference between the two groups regarding singleton pregnancies and preterm delivery or LBW.	Maximum score on the NOS
Schwarze et al. Chile, 2015 [28]	Retrospective cohort 2 years	ET (*n* = 6087) FET (*n* = 2123)	There was no significant difference between the two groups regarding singleton pregnancies and preterm delivery or LBW.	Maximum score on the NOS
Shavit et al. Canada, 2017 [29]	Retrospective cohort 4 years	ET (*n* = 575) FET (*n* = 161)	Singleton pregnancies following FET are associated with higher risk of macrosomia (*p* = 0.002). There was no significant difference regarding preterm birth, SGA, LBW or congenital malformations.	Maximum score on the NOS
Stormlund et al. Denmark, 2020 [30]	Randomized controlled trial, non-blinded, parallel 2 years, 4 months	ET (*n* = 66) FET (*n* = 61)	Fresh single blastocyst transfer led to an increased risk of preterm birth (*p* = 0.01), with no other differences observed regarding LBW, SGA or LGA.	Low risk of bias using the Cochrane Risk of Bias Tool
Zhang et al. China, 2018 [31]	Retrospective cohort 8 years, 2 months	ET (*n* = 2059) FET (*n* = 2053)	The FET group has significantly decreased risks of LBW (OR 0.59 [0.37–0.98]; *p* = 0.026) and SGA (OR 0.73 [0.55–0.99]; *p* = 0.041), but an increased risk of LGA (OR 1.26 [1.07–1.49]; *p* = 0.007) and macrosomia (OR 1.43 [1.16–1.75]; *p* = 0.001) in comparison with the ET group. There was no significant difference regarding congenital malformations.	Maximum score on the NOS
Zhang et al. China, 2020 [32]	Retrospective cohort 6 years	ET (*n* = 2125) FET (*n* = 924)	The incidence of macrosomia in the FET group was higher than in the ET group (OR 1.35 [1.07–1.71]; *p* = 0.013). Furthermore, FET singletons have a lower risk of LBW comparing to ET singletons (OR 0.67 [0.45–1.00]; *p* = 0.048). There was no significant difference between the two groups regarding preterm birth.	NOS score: S ★★★★ C ★★ O ★★★

* FET = frozen embryo transfer; ET = fresh embryo transfer; Preterm birth = delivery before 37 completed weeks of gestation; LBW = low birth weight (<2500 g); SGA = small for gestational age (<= −2SD); LGA = large for gestational age (>=+2SD); Neonatal death = death occurring in the first 28 days of life; NOS = Newcastle–Ottawa Quality Assessment Scale (maximum score on the NOS = 9 stars—4 stars for selection, 2 stars for comparability and 3 stars for outcome.

## Data Availability

Data are available upon request.

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
