# Peer review of "Fresh versus Frozen Embryo Transfer in In Vitro Fertilization/Intracytoplasmic Sperm Injection Cycles: A Systematic Review and Meta-Analysis of Neonatal Outcomes"

_medicina, 2024, doi:10.3390/medicina60081373_

Round 1

Reviewer 1 Report

Comments and Suggestions for Authors

Interesting meta-analysis on the neonatal outcomes following the transfer of fresh vs frozen embryos. the methodology is appropriate and the results are well presented. this is consistence with previous publications. Although not novel, it is interesting and acceptable for publication

Author Response

Thank you very much for reviewing our work and taking it into consideration for publication!

Reviewer 2 Report

Comments and Suggestions for Authors

The article is well written and complies with PRISMA standards. However, there are already several meta-analyses addressing this issue and the results reported in this and previous studies are similar. In the present study only 3 recent studies have been added in the analysis of the incidence of prematurity. Therefore, what is the justification for a new meta-analysis or update on this topic?

Other minor observations are as follows. 

* In Figure 1 there are asterisks in some boxes, but what they mean is not indicated in the figure caption. 

* Where are the results of the risk of bias assessment shown?

 it is required to indicate the bias of each study in a table or as a column of the table of characteristics of the included studies. 

* Publication bias analysis and sensitivity analysis are missing. 

Comments on the Quality of English Language

Minor editing of English language required.

Reviewer 3 Report

Comments and Suggestions for Authors

Comments to the manuscript medicina-3142641

This Systematic Review and Meta-Analysis provide an in-depth analysis of neonatal outcomes that occur after the transfer of fresh and frozen embryos (ET vs FET) in IVF/ICSI cycles. This topic is very important. But some paper has been published with this context.

1.          Roque M, Haahr T, Geber S, Esteves SC, Humaidan P. Fresh versus elective frozen embryo transfer in IVF/ICSI cycles: a systematic review and meta-analysis of reproductive outcomes. Hum Reprod Update (2019) 25:2–14. doi: 10.1093/humupd/dmy033

2.         Zaat T, Zagers M, Mol F, Goddijn M, Wely M, Mastenbroek S. Fresh versus frozen embryo transfers in assisted reproduction. Cochrane Database Syst Rev (2021) 2021:CD011184. doi: 10.1002/14651858.CD011184.pub3

etc

What is the novelty of your work compared to others?

The average age is very high, from 20 to 50 years old, with known causes of infertility. Could you please reduce or increase the age range?

20 cases is deficient? Can you add more, please

the authors should add a paragraph about the recommendations and perspectives

Comments on the Quality of English Language

Improvements should be made to the English language.

Reviewer 4 Report

Comments and Suggestions for Authors

The study follows the PRISMA guidelines and the protocol is registered at PROSPERO, which adds to the transparency and reliability.

The results are presented clearly with detailed descriptions of the included studies and their characteristics.

The references are up to date.

The introduction is comprehensive, but it could benefit from a brief mention of the clinical significance of the study's findings.

Author Response

Thank you very much for reviewing our work and for the insightful comments, we will take them into consideration when revising the manuscript.

Round 2

Reviewer 2 Report

Comments and Suggestions for Authors

The authors have introduced the suggested changes and have extensively justified the importance of their article. 

Author Response

Thank you very much for your insightful suggestions, they have been a definite improvement to our manuscript!

Reviewer 3 Report

Comments and Suggestions for Authors

Fresh versus frozen embryo transfer in IVF/ICSI cycles: a systematic review and meta-analysis of neonatal outcomes

Comments to the manuscript medicina-3142641-R1

The authors have made some corrections. However, the authors should increase the size of the cases because statistically 20 is not representative. In addition, the authors should add paragraphs about Recommendations and Perspectives.

Thanks

Comments on the Quality of English Language

Minor editing of English language required
